# Drifting discrete Jovian radio bursts reveal acceleration processes related to Ganymede and the main aurora

Emilie Mauduit [1,5], Philippe Zarka [1,2,5] ✉, Laurent Lamy [1,2,3] & Sébastien L. G. Hess[4]

Radio detection at high time-frequency resolutions is a powerful means of remotely studying electron acceleration processes. Radio bursts have characteristics (polarization, drift, periodicity) making them easier to detect than slowly variable emissions. They are not uncommon in solar system planetary magnetospheres, the powerful Jovian "short bursts (S-bursts)" induced by the Io-Jupiter interaction being especially well-documented. Here we present a detection method of drifting radio bursts in terabytes of high resolution time-frequency data, applied to one month of ground-based Jupiter observations. Beyond the expected Io-Jupiter S-bursts, we find decameter S-bursts related to the Ganymede-Jupiter interaction and the main Jovian aurora, revealing ubiquitous Alfvénic electron acceleration in Jupiter's high-latitude regions. Our observations show accelerated electron energies are distributed in two populations, kilo-electron-Volts and hundreds of electron-Volts. This detection technique may help characterizing inaccessible astrophysical sources such as exoplanets.

Acceleration of charged particles well beyond their thermal speed and subsequent production of non-thermal electromagnetic emissions is at the origin of various transient phenomena in astrophysics, such as solar or stellar flares, and planetary or exoplanetary radio bursts (see e.g. refs. 1–7). Planetary magnetospheres are unique laboratories where such acceleration processes can be studied in situ by spacecraft, while the resulting electromagnetic emissions can be detected remotely. This is the case in particular for auroral and satellite-induced emissions, produced by electrons accelerated to energies of a few keV to hundreds of keV[8,9]. Electrons with energies up to a few tens of keV generate Electron Cyclotron Maser (ECM) radio emissions due to their gyration around magnetic field lines[10], while electrons with energies up to a few hundreds of keV give rise to UV emissions after they have collided with atmospheric atoms at the footprints of these field lines (see e.g. refs. 11,12). Ubiquitous ECM radio emissions from solar system

planets display slowly varying components (minutes to hours) and fine structures (sub-second bursts)[10,13]. These bursts are emitted at frequencies that are related to the altitude of the sources, hence transport detailed information of the motion of these sources, as well as of the processes accelerating the source electrons. At Earth, auroral kilometric radiation displays fine structures drifting positively and/or negatively in the $t–f$ plane, consistent with shell-driven ECM excited in drifting electrostatic structures[14–18]. There, the main electron acceleration process involves parallel potential drops along magnetic field lines[19], but Alfvén waves have also been invoked in some cases[20]. At Jupiter, the decameter-wave emission called S-bursts related to the Io-Jupiter interaction are remarkable discrete, quasi-periodic features, which drift mainly negatively in the $t–f$ plane (from high to low frequencies, at −10 to −30 MHz/s, in a few tens to a few hundred msec)[21]. These bursts have motivated several modeling efforts that, combined

[1]LESIA, Observatoire de Paris, Université PSL, CNRS, Sorbonne Université, Université de Paris, 5 place Jules Janssen, Meudon 92195, France. [2]Observatoire Radioastronomique de Nançay (ORN), Observatoire de Paris, CNRS, PSL, Université d'Orléans, OSUC, route de Souesmes, Nançay 18330, France. [3]LAM, Pythéas, Pôle Etoile, Aix Marseille Univ, CNRS, CNES, Site de Château Gombert, 13388 Marseille, France. [4]DPHY, ONERA, Université de Toulouse, 2 Avenue Edouard Belin, 31055 Toulouse, France. [5]These authors contributed equally: Emilie Mauduit and Philippe Zarka. ✉e-mail: philippe.zarka@obspm.fr

together, provide a coherent generation scenario : the interaction of Io with Jupiter's magnetic field is known to excite Alfvén waves, which propagate along the Io Flux Tube (IFT)[22,23]; at the base of the IFT footprints these waves enter the ionospheric Alfvén resonator, a region close to Jupiter's ionosphere where the Alfvén velocity drops abruptly[24,25]; this resonant cavity can amplify Alfvén waves at specific frequencies related to the properties of the cavity (in the range 10−20 Hz at Jupiter)[25,26]; these waves give rise to parallel bi-directional electric fields over a broad range of altitudes along the IFT ; at the lower edge of this range, 0.5−1 $R_J$ (= Jupiter's radius = 71492 km) above Jupiter's ionosphere, these parallel electric fields were shown to be able to accelerate downwards electrons up to 1−10 keV over half an Alfvén wavelength[27]. Following acceleration, the evolution of the electron's distribution function was computed versus time and position along the IFT, and found to be modulated at the period of the Alfvén wave injected in the simulation[27]; then loss-cone driven ECM emission growth rates were calculated, in order to finally produce synthetic radio dynamic spectra showing quasi-periodic discrete bursts drifting in the time-frequency plane at several MHz/s, very similar to the observed ones[27,28]. Bursts discreteness is due to the acceleration pulse associated with each Alfvén wavelength, whose periodicity (about 5−20 Hz) is imprinted in the subsequent radio emission[25–27]. Bursts drift results from the adiabatic motion of accelerated electrons bunches along Jovian magnetic field lines. Until now, S-bursts have been observed only associated with the Io-Jupiter interaction[21]. But they should exist wherever Alfvénic electron acceleration and the Alfvén resonator are present.

Here we develop a detection method for drifting bursts adapted to an application on massive high $t$–$f$ resolution data, and we show the first blind search for fast drifting radio bursts in ground-based radio observations of Jupiter, the only ones that can provide high enough temporal and spectral resolutions. With this method, we show the existence of S-bursts associated with the Ganymede-Jupiter interaction and the main Jovian aurora. This reveals the ubiquitous character of Alfvénic electron acceleration and of the Alfvén resonator in Jupiter's high-latitude regions. We estimate the Alfvén wave periods and the energy of accelerated electrons. Two populations are found to co-exist, with different energies (a few keV and a few hundred eV).

## Results

We have used the Nançay Decameter Array (NDA), which has been dedicated for more than 4 decades to the daily monitoring of Jupiter's low-frequency (auroral and satellite-induced) radio emissions[29–31]. Since mid-2016, a high time-frequency resolution spectrograph called JunoN (for Juno-Nançay) operates at the NDA in support to the Juno mission. It records left-hand (LH) and right-hand (RH) circularly polarized flux densities at resolutions up to 2.6 ms × 3.05 kHz in selected time intervals (especially within ± 1–2 days of Juno perijoves)[31]. Here we have analyzed a total of 24 hours of high-resolution JunoN data recorded over one month (April 2021), representing approximately 67 million spectra over 16384 frequency channels each, and a data volume > 4 Terabytes. The data have been sliced in approximately 160000 dynamic spectrograms covering each 1.1 s × 9 MHz, in which drifting bursts show up after mitigation of radio frequency interference (RFI). As it is impossible to analyze visually such a large number of time-frequency images, we have developed an automated pipeline that detects the presence of drifting features in a dynamic spectrum and measures their drift rate and signal-to-noise ratio (SNR). The pipeline, which includes several steps (RFI mitigation, 2D FFT followed by Radon transform, and Gaussian fit of the dominant peak in the Radon spectrum) is described in details in the Methods section.

With it, we have detected fast drifting fine structures in 24% of the RH dynamic spectra (corresponding to emissions from Jupiter's northern hemisphere) and 18% of the LH ones (southern hemisphere) (Table 1). These high percentages result from the pre-selection, by the NDA operator at the end of each JunoN observation, of data intervals potentially containing fine structures. Relative to the total observation time of Jupiter, these percentages are of order of 2–3%, consistent with previous estimates[32].

We have identified the origin of the detected bursts on a statistical basis. The Io-Jupiter interaction was first demonstrated by plotting the occurrence of the decameter radio emission versus the Jovian longitude of the observer (CML) and the orbital phase of Io[33]. The gray-shaded map in Fig. 1a is the modern version of this occurrence map, based on 26 years of NDA observations[34]. Io-induced emissions correspond to the enhanced occurrence patches that depend on the coordinate "Io Phase" (identified by white boxes labeled with the usual conventions applied to Jovian radio components[34]), while the vertical bands varying only with the CML are called "Io-independent" or "non-Io" emissions. The less prominent Ganymede-induced decameter emissions were statistically discovered and characterized by plotting the occurrence of non-Io emissions versus the CML and the phase of Ganymede[35] (gray-shaded map in Fig. 1b, where the white boxes identify enhanced occurrence patches that depend on the Ganymede Phase). Still weaker Europa-induced emissions were identified similarly by plotting the occurrence of non-Io and non-Ganymede emissions versus the CML and the phase of Europa[36]. Non-satellite emissions depending on the CML only are related to Jupiter's main aurora. In Fig. 1, our burst detections were overplotted in color on top of the CML −Io phase (Fig. 1a) and CML−Ganymede phase (Fig. 1b) maps.

When a detection falls into one of the boxes of Fig. 1a (respectively Fig. 1b), it is very likely that the corresponding drifting structure is produced in the IFT (resp. in the Ganymede flux tube), with a probability of 60–80%, equal to $(Occ_{in}−Occ_{out})/Occ_{in}$ with $Occ_{in}$ the average occurrence level in the box and $Occ_{out}$ that around the box. For a cluster of $N$ points in a box, the probability that none of them is related to the corresponding satellite falls rapidly to zero as $(Occ_{out}/Occ_{in})^N$. Figure 1 reveals thus that fine drifting structures are generated in the IFT (blue symbols), but also in the Ganymede flux tube (green symbols) and above the main Jovian auroras (pink symbols - we checked that those are not related to Europa either).

Figure 2a, b shows two examples of a series of drifting bursts related to the IFT, with a clear quasi-periodic structure (period about 26 ms) and a drift rate (slope) of about −13 MHz/s measured around a frequency of 15 MHz. Figure 3a, b shows the distribution of slopes for all our detected IFT-related structures, in RH (resp. LH) polarization, corresponding to radio bursts originating from Jupiter's northern (respectively southern) IFT footprints[10]. The vast majority of slopes lies between −12 and −24 MHz/s (with a tail down to −30 MHz/s), consistent with earlier works[21,37]. For a radio emission at the local electron cyclotron frequency ($f_{ce}$), according to Jupiter's internal magnetic field

## Table 1 | Number of detections as a function of their likely origin and their drift rate

| Origin | Polarization | df/dt < − 10 MHz/s | − 10≤df/dt < 0 MHz/s | df/dt ≥ 0 MHz/s | Total |
|--------|-------------|-----------|-----------|-----------|-------|
| Io | LH | 10560 | 790 | 167 | 11517 |
|    | RH | 14492 | 1126 | 238 | 15856 |
| Ganymede | LH | 357 | 580 | 270 | 1207 |
|          | RH | 52 | 155 | 286 | 493 |
| Main aurora | LH | 429 | 601 | 626 | 1656 |
|             | RH | 664 | 1680 | 558 | 2902 |
| Total | LH | 11346 | 1971 | 1063 | 14380 |
|       | RH | 15208 | 2961 | 1082 | 19251 |

We show in this table the number of 1.1 s × 9 MHz dynamic spectra (out of approximately 80 000 processed for each polarization) in which fast drifting structures have been detected, as a function of their likely origin and polarization (Fig. 1) and drift rate, df/dt (Fig. 3).

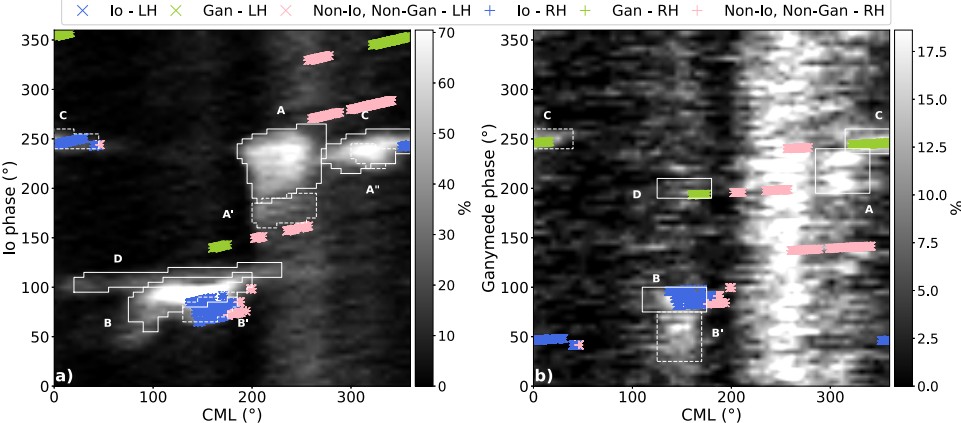

**Fig. 1 | Occurrence of detected S-bursts (a) as a function of the CML (Central Meridian Longitude = sub-observer's longitude) and orbital phase of Io, and (b) as a function of the CML and orbital phase of Ganymede.** The background gray-shaded maps represent the occurrence probabilities of all Jovian decameter emissions (**a**) and of non-Io decameter emissions only (**b**), reproduced from ref. 35 ("Reproduced with permission from Astronomy & Astrophysics, © ESO"). Over-plotted colored symbols corresponds to our burst detections. The same series of symbols are displayed on both panels, with same CML but different satellite phases.

Symbols included in Io-induced white boxes in (**a**) are displayed in blue (some of them are also included in Ganymede-induced boxes in (**b**) due to the 4:1 orbital resonance between Io and Ganymede). Symbols matching Ganymede-induced boxes are displayed in green. The remaining symbols, displayed in pink, correspond to radio bursts unrelated to Jupiter's moons (we checked that they are not related to Europa either) and thus produced above or close to the Jovian main auroral oval.

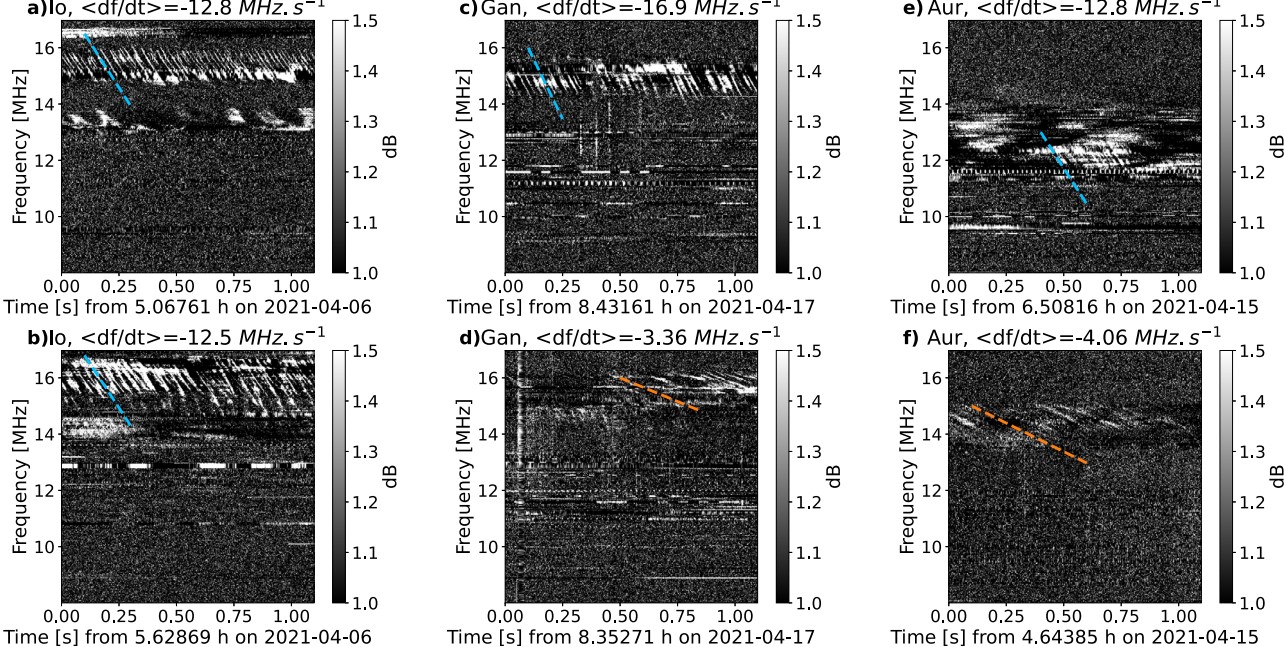

**Fig. 2 | Examples of NDA/JunoN dynamic spectra illustrating the detection of fast drifting bursts of various origins.** Panels (**a**) and (**b**) show examples of S-bursts related to Io. Panels (**c**) and (**d**) show examples of fine structures related to Ganymede, with two types of drifts. Finally, panels (**e**) and (**f**) show examples of fine structures related to the main aurora, also with two types of drifts. Average drift rates are indicated as color lines on each plot (with the same color code as in Fig. 3, see below). A blue dashed line corresponds to a drift rate lower than −10 MHz/s, and an orange one to a drift rate between −10 and 0 MHz/s.

models (see e.g. ref. 38), the decameter radio sources at 10–40 MHz are located at a radial distance of 1 to 1.5 Jovian radius from Jupiter's center, and a drift rate of 1 kHz/s corresponds to a source velocity of 1–2 km/s parallel to the magnetic field lines. For bunches of emitting electrons, drift rates of −12 to −24 MHz/s thus imply a parallel kinetic energy of 0.5–8 keV and a total kinetic energy of 1.3 to 13 keV for individual electrons (see Methods, subsection Drift rates and electron energies). A small fraction of slopes is also detected between −3 and −10 MHz/s, which correspond to the slow-drift bursts noted in refs. 39,40, without a clear explanation of their origin. Finally, we have checked that the small fraction of positive slopes all correspond to

spurious detections of weak residual RFI with a complex time-frequency structure (see Supplementary information, Section 1).

Figure 2c, d displays drifting bursts related to the Ganymede flux tube, also showing clear quasi-periodic structures. Here, slopes about −17 MHz/s (with a period of about 31 ms) and −3.4 MHz/s (with a period about 38 ms) are measured around a frequency of 15 MHz. Figure 3c, d shows that, as for IFT bursts, many slopes lie between −14 and −22 MHz/s (with a tail down to -30 MHz/s). However, in contrast with IFT bursts, an equivalent number of slopes are detected between −3 and −7 MHz/s (as well as spurious positive slopes). These two families of drift rates correspond to total electron energies of 2–10 keV and

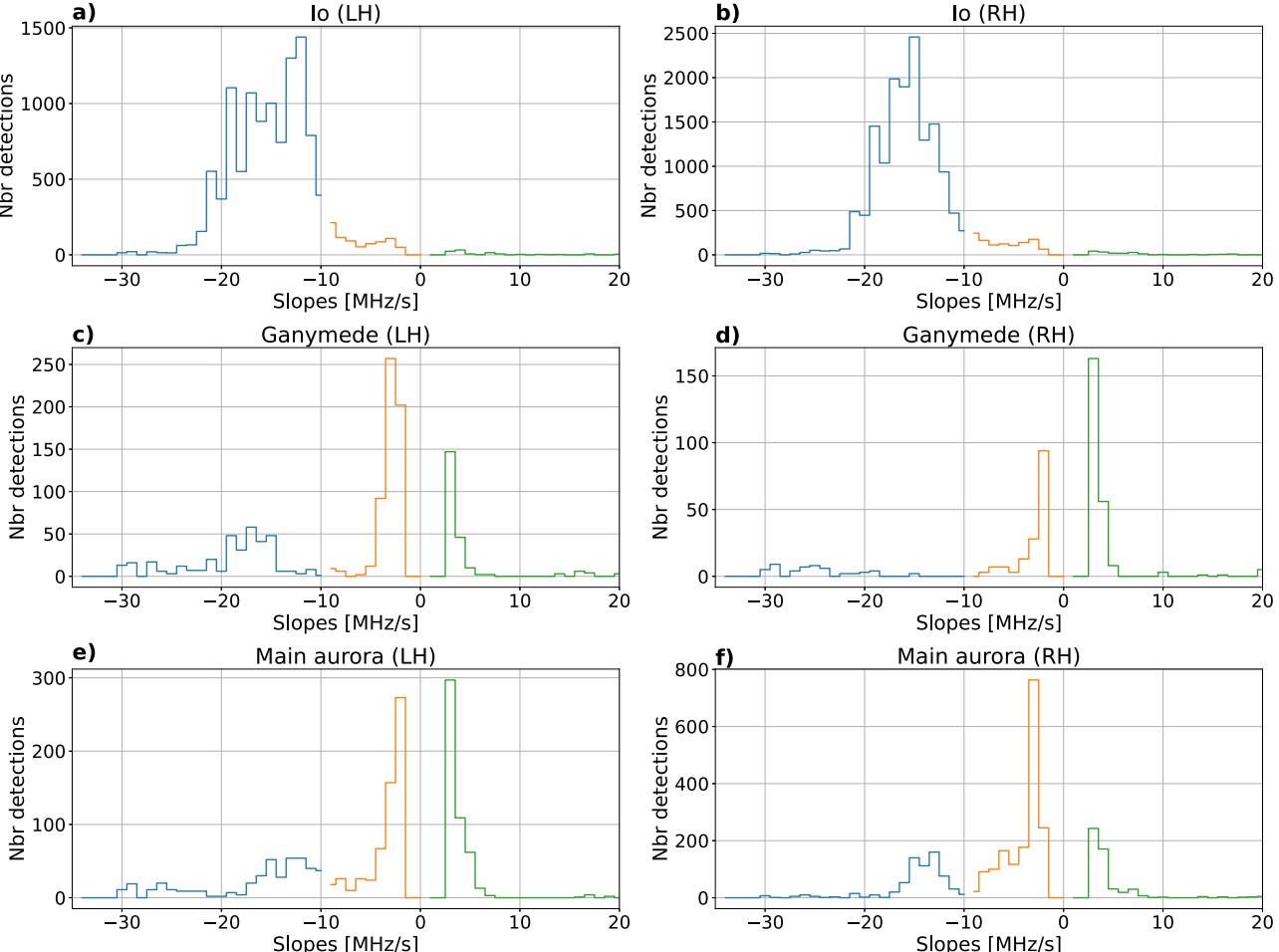

**Fig. 3 | Histograms of slopes (drift rates) in LH polarization (first column) and RH polarization (second column).** In panels (**a**) and (**b**) that correspond to Io-induced emissions, most of the measured drifts lie around −15 MHz/s (blue color), as previously known. In panels (**c**) and (**d**), which concern Ganymede-induced bursts, a relatively smaller peak is observed for the same drifts plus a second peak around −4 MHz/s (orange). Similar results are observed for the auroral (satellite-independent) drifting bursts in panels (**e**) and (**f**). Positive slopes (green) are due to residual RFI (see Supplementary information, Section 1). In Table 1, we summarize the number of detections associated with each slope interval.

approximately 0.1–1 keV, respectively (see Methods, subsection Drift rates and electron energies). Bursts with lower absolute drift rates are generally less intense than fast-drifting bursts. Finally, for drifting features related to the satellite-independent (auroral) radio emissions, we find characteristics similar to Ganymede-induced radio emissions, but with slightly slower drift rates and thus lower electron energies, as illustrated in Figs. 2e, f, 3e, f and subsection Drift rates and electron energies.

## Discussion

One puzzling result is the fact that for Ganymede-induced and auroral bursts, we observe comparable numbers of slow (about −5 MHz/s, i.e. drifting electrons < 1 keV) and fast (about −15 MHz/s, i.e. >2 keV electrons) drift rates, whereas the latter are very largely dominant for Io-induced bursts. Why the slower drifting (lower energy) population is much more prominent for Ganymede-induced and auroral bursts than for Io-induced bursts ? A related question is why do they form relatively distinct distributions rather than a continuum ? The response may invoke several mechanisms of excitation of Alfvén waves (the low-altitude Alfvén resonator[25,26], Ganymede-Jupiter magnetic reconnection[41], magnetosphere-ionosphere coupling, solar wind-magnetosphere interaction[42], field line resonances[43]), time variations in their efficiency[44] or a different process generating drifting sources (see discussion in the next paragraph).

The mere existence of bursts with slopes about −5 MHz/s raises the question of their generation. While energies of 2–10 keV are typical for electrons generating Jupiter's high latitude radio emissions on the extraordinary (X) magneto-ionic mode via the ECM mechanism (see e.g. refs. 10,27,37,45), energies of <1 keV are much less often encountered in this context. In the presence of a background of thermal electrons, hot electrons distributions with a characteristic energy <0.5 keV do not lead to significant ECM growth rates[46]. Positive growth rates are obtained for the X mode second harmonic and for the ordinary (O) mode. There is no observational reason for the slowly drifting bursts to be second harmonic emission (although it is not excluded, as the fundamental would lie below the low-frequency limit of our observations), but the possibility of O mode emission, consistent with the weaker intensity of slowly drifting bursts, should be investigated via statistics on the polarization of the emissions. Supplementary Fig. 4 in Supplementary information indeed supports the presence of some O mode emission. Another possibility is that the thermal plasma is absent and the source is dominated by hot electrons. In that case, fundamental X mode ECM emission driven by <1 keV electrons might be possible[45,47]. From the elliptical polarization of Jupiter's decameter radio emission, it was proposed that the density in and close to the sources should be very small ($n_e \leq 5\,\mathrm{cm}^{-3}$, implying $f_{pe}/f_{ce} \leq 10^{-3}$, with $f_{pe}$ the plasma frequency)[48]. Such low densities are indeed observed by Juno across the entire auroral zone[9]. Finally one cannot exclude the

possibility that some drift rates are not related to the quasi-adiabatic motion of emitting electrons, but rather to the motion of the source itself[39,40,49]. Drifting potential drops such as electron or ions holes have been identified by refs. 28,50, but with a much lower velocity (5–15 km/s).

In summary, we have developed a robust automated detection method for drifting discrete radio bursts. The method has been tested successfully by detecting the well-known drifting radio bursts from the Io-Jupiter circuit, with their known slopes. Then we have discovered similar quasi-periodic, fast-drifting radio bursts associated to the Ganymede-Jupiter interaction and to the main Jovian auroras. These results suggest that Alfvénic acceleration is ubiquitous in Jupiter's auroral regions, albeit not necessarily the dominant mechanism (whistler waves and large-amplitude solitary electric field structures, both observed at Jupiter[9,51] are also good candidates). They are consistent with the magnetic fluctuation levels and broadband electron energy spectra detected by Juno along its polar orbit and low-altitude perijoves. In the magnetic flux tubes connected to Io's UV footprint and tail, both transverse magnetic perturbations and broadband electron energy spectra (up to a few 10s keV) have been observed[52–55], consistent with acceleration of electrons by the parallel electric fields of the Alfvén waves that are known to transport Io's perturbation of the Jovian magnetic field down to the northern and southern ionospheres of the planet[22,23,56–58]. A similar situation exists at Ganymede / in the Ganymede flux tube[53,59,60]. High levels of magnetic fluctuations identified as Alfvén waves have also been observed along auroral field lines above the diffuse region equatorward of the main aurora, where 3-30 keV electrons are present[9,53,61], and where Jupiter's auroral radio sources have been located[62,63]. Our measurements point to Alfvén waves with 20–45 ms periods (22–50 Hz frequencies), and support the ubiquitous character of the Alfvén resonator in Jupiter's high-latitude regions. Parallel potential drops also exist, e.g. along the IFT[28,37,50], but they appear secondary to Alfvén waves for bulk electron acceleration.

We have demonstrated that our detection method opens the possibility to remotely study acceleration processes using ground-based high-resolution radio measurements. The next step will be to apply it to all data recorded at high time-frequency resolution with the NDA/JunoN receiver since 2016 (about 100 TB), and study statistically the occurrence (and its time variations), drift rates (depending on the frequency) and polarization of the detected bursts. Beyond that, there is great general interest in exoplanets, and radio observations have played an important role in deducing magnetic fields on such remote objects as brown dwarfs[64–67], which, with present technologies, is the closest thing to an exoplanet that can likely be detected right now. Radio appears presently more adapted than UV or infrared for detecting and monitoring the activity of astrophysical sources like exoplanets, star-planet interactions and ultracool dwarfs[1,68–71]. Radio observations can give access to higher time-frequency resolutions, revealing sporadic and or narrowband bursts that provide constraints on their generation mechanism as well as on electron acceleration[72]. Detection of drifting radio bursts is also interesting at low frequencies because it is less hindered by the intense sky background, RFI, and ionospheric propagation effects that make the detection of slowly varying signal very difficult[73]. The method here has thus promising astrophysical applications.

## Methods
### Data analyzed
The data used in this study have been recorded with the Juno-Nançay receiver operating at the NDA in support of the Juno space mission[74]. NDA/JunoN observes Jupiter a few hours per day and produces dynamic spectra of LH and RH flux densities at a resolution of 2.6 ms × 3.05 kHz, totaling about 3 TB per day. Such a data volume cannot be stored in its entirety at this resolution thus, after each observation, the NDA operator examines the recorded dynamic spectra at a

degraded resolution (83 ms × 12 kHz) and identifies intervals containing Jovian radio emission and, among these, sub-intervals containing fine time-frequency structures. The intervals containing only slow Jupiter radio signals are stored at the degraded resolution (83 ms × 12 kHz) whereas, when bursts are thought to be present, the data is stored at its original resolution. This visual identification is only a first coarse selection: the minimum duration of high resolution intervals stored is about 20 min, but these intervals may actually contain shorter sequences of fast drifting bursts separated by intervals containing smooth or no emission. This is why we developed a method that automatically detects the presence of discrete fast drifting radio bursts in a dynamic spectrum, amidst superimposed RFI that are frequent and prominent at low frequencies, and that measures their slopes. The algorithm was first tested and optimized on simulated signals, except for the RFI mitigation that has been optimized on real data. The pipeline was then applied to one month of data (april 2021) for this analysis. Since the structures we are looking for may come and go on short timescales, we have processed the data by chunks of 1.1 s × 9 MHz in both LH and RH circular polarizations. Figure 4 summarizes the processing steps detailed below.

### Processing method
**RFI mitigation.** The first step is to remove fixed frequency RFI (e.g. broadcast stations) and broadband sporadic RFI (e.g. lightning), i.e. horizontal and vertical lines on dynamic spectra, without corrupting Jupiter signals which can also be very intense. For that purpose, we integrate the dynamic spectrum over all frequencies to obtain a 1D time series, and over all time steps to obtain a 1D average spectrum. Each of these 1D curves is iteratively examined to identify outliers exceeding a given threshold ($3.5\sigma$). The corresponding data is flagged and interpolated from surrounding unflagged pixels. This procedure is applied first on data expressed in decibels (the logarithmic scale compresses the dynamic range and allows to identify the strongest RFI). The resulting dynamic spectrum is flattened through division by a background (average) spectrum. Then the above procedure is applied again, this time on data in linear scale (to identify weaker RFI). This processing step allows us to transform Fig. 4a into Fig. 4b.

**2D-fast Fourier transform.** Then the 2D-FFT of the cleaned dynamic spectrum resulting from step 1 is computed (Fig. 4c). This transform gathers the energy associated with fine structures with a given orientation in the $(t, f)$ plane into a single line in the plane $(t^{-1}, f^{-1})$, centered on $(0, 0)$ and perpendicular to the slope in the $(t, f)$ plane. Residual RFI not eliminated in step 1 appears thus as a vertical band and a horizontal band in Fig. 4c. The contrast of this figure is enhanced by dividing each line by the 1D average over $f^{-1}$ and each column by the 1D average over $t^{-1}$. This procedure results in Fig. 4d.

**Radon transform.** Next, the Radon transform of the enhanced centered 2D-FFT is computed. It consists of integrating the values in Fig. 4d along a rotating line centered in $(0, 0)$, in the direct sense from the -y axis. This results in a distribution of intensities with respect to the angle between the rotating line and the vertical axis $I(\alpha)$, with $\alpha \in [0°, 180°]$, as displayed in Fig. 4e (green line). This use of the Radon transform was introduced in ref. 75 for real-time detection of drifting fine structures. It was applied to the analysis of high-resolution dynamic spectra recorded at the Kharkiv/UTR-2 radiotelescope in ref. 50. The variable length of the rotating line along which the 2D-FFT is integrated leads to a double-peak modulation of the Radon transform at angles $\alpha = 45°$ and $\alpha = 135°$ (due to the square shape of the 2D-FFT). This bias was eliminated by ref. 50 via the computation of an ad-hoc contrast $C(\alpha) = I(\alpha)/I(\alpha + 90°) - 1$. With our simulations, we have found that a better correction of this bias consists of dividing $I(\alpha)$ by the Radon transform of a uniform square of the same size as Fig. 4d, which we note $I_1$ (orange dashed line in Fig. 4e). Finally, in order to

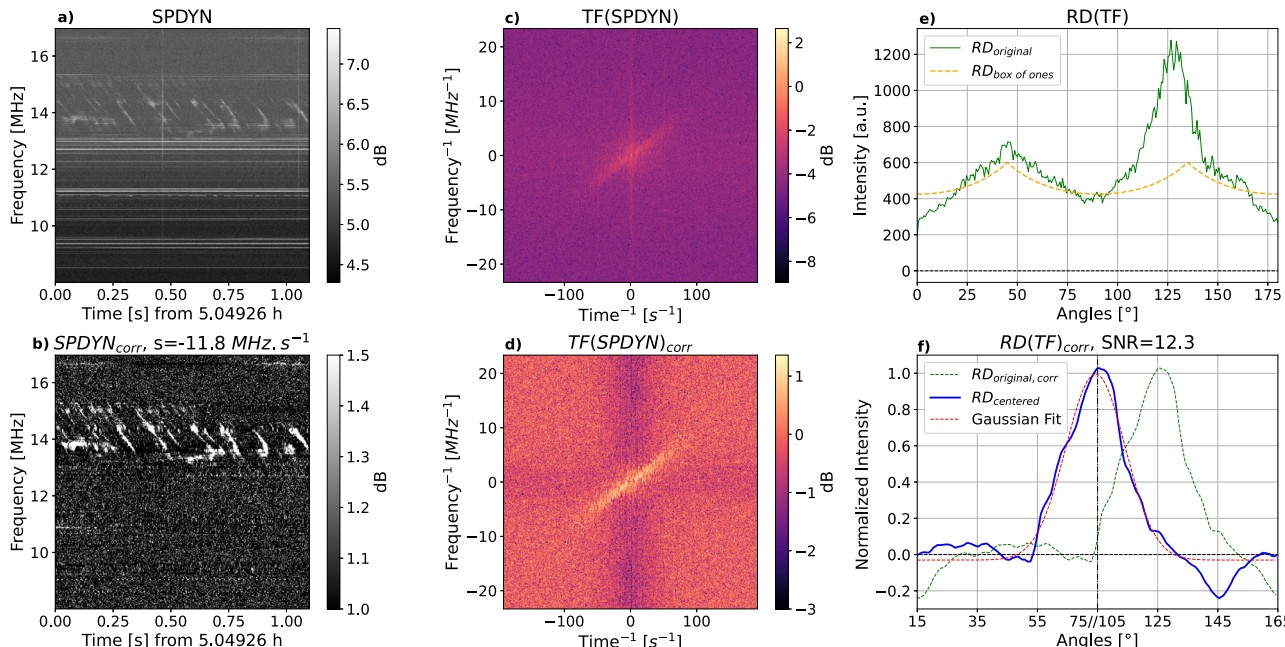

**Fig. 4 | Detail of the processing steps applied to each data chunk.** Panel (**a**) shows the original 1.1 s × 9 MHz dynamic spectrum. Panel (**b**) shows the same data after RFI-mitigation and background subtraction, which enhance the SNR. Panel (**c**) displays the 2D-FFT of panel (b) data, and panel (**d**) is the corrected 2D-FFT (divided by its integrated value along each axis), where the inclined centered feature corresponds to the drifting bursts signal. Panel (**e**) is the result of the Radon transform applied to panel (**d**) (green solid line), as well as the Radon transform of a uniform square of the same size as panel (**d**) (orange dashed line). Finally, panel (**f**) shows the corrected and normalized Radon transform (or spectrum−dotted green) as a function of the angle of the rotating diameter, after excluding intervals 0° ± 15° and 90° ± 15°. This spectrum, centered (blue line) and fitted by a Gaussian curve (dotted red) provides the detection, its SNR, and the slope of the detected structures.

obtain a flat background centered on zero, we computed the following function, which is displayed as the green dotted line in Fig. 4f:

$$C(\alpha) = \frac{I(\alpha)/I_1}{\text{median}(I(\alpha)/I_1)} - 1 \qquad (1)$$

**Peak detection.** We want to detect automatically the main peak in $C(\alpha)$, with its SNR and angle. To avoid pollution by any RFI that may subsist after the previous processing steps, we exclude the intervals $\alpha = 0° \pm 15°$ (i.e. $\alpha \in [0°, 15°]$ and $[165°, 180°]$), and $\alpha = 90° \pm 15°$. The two remaining intervals of $\alpha$ are displayed side by side, covering a range of 120° in total as in Fig. 4f. Note that the value of $\alpha$ in abscissa is discontinuous, jumping from 75° to 105° at the middle of the x-axis. The maximum value of $C(\alpha)$ is then located and centered (as illustrated by the solid blue line in Fig. 4f), and it is fitted by a Gaussian function (red dotted line in Fig. 4f), from which various parameters (slope $\alpha$, amplitude, SNR, $1\sigma$ − width) are retrieved and used for characterizing the drifting bursts. The SNR that is used to detect drifting fine structures above a given threshold is the amplitude of the fitted Gaussian function divided by the standard deviation of the noise outside ($\pm 2\sigma$) of its peak.

Figure 5 shows an example of the result obtained when no Jupiter signal is present in the dynamic spectrum. The method still finds and fits a peak, but the low associated SNR ensures that we can conclude that there is no detection.

**Spectral integration before processing.** We want our search for drifting bursts to cover a broad range of drift rates, typically $5 \leq |df/dt| \leq 25$ MHz/s. Known drift rates of Io-Jupiter S-bursts are in the range −15 to −25 MHz/s[21,40]. The Radon transform performs better far from the horizontal or vertical lines that may be polluted by RFI, i.e. in the range $45° \pm 25°$. At the full resolution of the JunoN receiver, an angle of 45° corresponds to a drift rate $|\delta f/\delta t| = 3.05 \times 10^{-3}/2.6 \times 10^{-3} = 1.2$ MHz/s. A burst drifting at -20 MHz/s makes an angle $\alpha$ at 3.3° from the vertical axis in the dynamic spectrum, and thus has all chances of being

considered as an RFI. The spectral resolution of JunoN is thus comparatively too high with respect to its time resolution. To correct for that, we pre-integrated the data by a factor 7 in frequency, bringing thus the spectral resolution to 21.35 kHz. With resolutions of 2.6 ms × 21.35 kHz, the range $20° \leq \alpha \leq 70°$ corresponds to drift rates $|\delta f/(\delta t \cdot \tan \alpha)| \simeq 3$–23 MHz/s, well adapted to our search.

**Processing pipeline and outputs.** Our automated processing pipeline combines data reading with all the above processing steps, and produces a vector of results for each processed data chunk. We are interested in the spectral range from about 10 MHz (ionospheric cutoff) to 40 MHz (upper limit of Jupiter's decameter emissions). Within this range, the drift rate of S-bursts varies due to the combined adiabatic change of the parallel velocity of the electrons and to the magnetic field topology[21]. Thus we decided to split this spectral range into four bands of approximately 9 MHz each (actually 9074 kHz), with 1 MHz overlap at their edges: 8–17, 16–25, 24–33, and 32–41 MHz. Each band gathers 9074 kHz/21.35 kHz = 425 channels. Square dynamic spectra of 425 × 425 pixels thus last for 425 × 2.6 ms = 1.1 s, a duration well adapted to the sporadicity of Jupiter's bursts.

Data is thus read by groups of 425 spectra (1.1 s) × 2 polarizations (LH and RH), which are first rebinned by a factor 7 in frequency. Eight dynamic spectra of 425 × 425 pixels are then extracted corresponding to the above four spectral bands × 2 polarizations, and each of them goes successively through all above processing steps. A vector of results is obtained (and stored) for each processed 1.1 s × 9 MHz dynamic spectrum, which contains the following information:

- *ifile*: the index of the file being processed,
- *tag*: an integer indicating the presence (=1) or absence (=0) of fine drifting structures in the processed dynamic spectrum,
- *iband,ichunk*: the index spectral band and time band of the processed dynamic spectrum,
- $I_{max}$, $err(I_{max})$: the peak value of the fitted Gaussian and the error on its estimation,

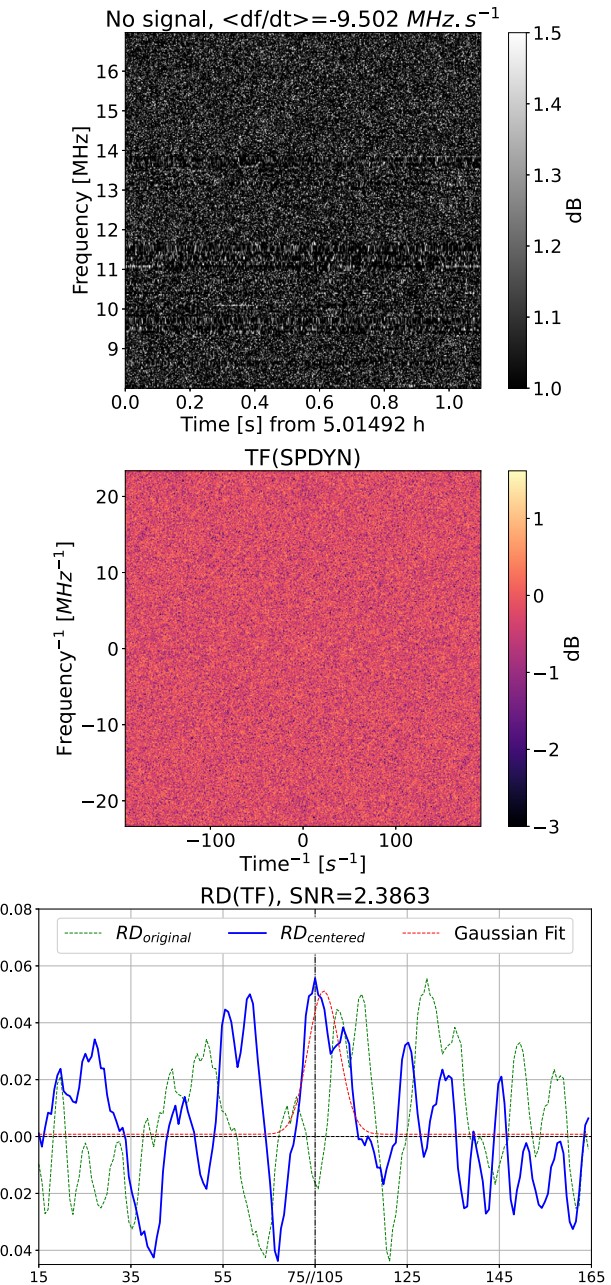

**Fig. 5 | Example of non-detection of fine drifting structures.** There is no Jupiter signal in the processed dynamic spectrum (top), thus the 2D-FFT is simply noise (center) and the resulting Radon transform does not have any significant peak (bottom). A threshold on the SNR (usually 5) allows us to conclude to non-detection in this case.

- $\alpha, err(\alpha)$: the angle corresponding to $I_{max}$ and the error on its estimation,
- $\sigma, err(\sigma)$: the width (standard deviation) of the fitted Gaussian and the error on its estimation,
- $\chi^2$: the quality of the fit,
- $SNR$: the SNR value computed as $I_{max}$ divided by the standard deviation of the noise outside of $\alpha \pm 2\sigma$,
- $err(RD)$: the mean error between the Radon transform and the fit,
- $tmin, tmax$: the starting and ending times of the processed dynamic spectrum,
- $fmin, fmax$: the minimum and maximum frequencies of the processed dynamic spectrum.

An example of such vectors are provided as a Supplementary Data file.

These results are saved for subsequent statistical analysis. We have set our detection criterion at an SNR of 6, thus SNR ≥ 6 → tag = 1. When fine drifting structures are detected, the dynamic spectrum, its 2D-FFT and Radon transform are plotted and saved in a pdf file, which allows us afterwards to check the reality of any detection.

**Drift rates and electron energies.** The conversion of observed drift rates in the $t$–$f$ plane into electron energies requires a description of the topology of the Jovian magnetic field at the source location. As the emission is produced at $f \simeq f_{ce} = eB/2\pi m$ (with $e$ the charge of the electron and $m$ its mass), the drift rate $df/dt$ is proportional to $dB/dt$, which can be written:

$$\frac{dB}{dt} = \frac{dB}{ds} \times \frac{ds}{dt} = \frac{dB}{d\theta} \times \frac{d\theta}{ds} \times v_{//} \qquad (2)$$

with $s$ the curvilinear abscissa along the source field line, $\theta$ the colatitude and $v_{//}$ the parallel velocity of the emitting electrons. In the general case, the source field line must be determined from the time of observation and some assumption or determination of the radio beaming angle (see e.g. ref. 44). Then the function $dB/ds(f)$ can be computed numerically from the magnetic field topology, given by a multipolar model such as ref. 38. When studying the distribution of many measurements of $df/dt(f)$, corresponding to radio bursts generated on many field lines, assuming a dipolar Jovian magnetic field allows us to compute analytically the corresponding distribution of parallel and total electron velocities and thus energies. With this assumption, the field amplitude writes:

$$B(R, \theta) = \frac{B_e}{R^3}(1 + 3\cos^2\theta)^{1/2} \qquad (3)$$

with $B_e$ the surface equatorial magnetic field amplitude and $R$ the radial distance in $R_J$. Following ref. 21, the dipolar moment is taken equal to 7 $G \cdot R_J^3$ (instead of 4.2 $G \cdot R_J^3$ in multipolar field models[38]) in order to reach cyclotron frequencies of 40 MHz near the pole and thus compensate for neglecting mutipolar terms. $B_e$ is thus equal to $7G$. The equation of a (plane, meridian) dipolar field line is:

$$R = L \sin^2\theta \qquad (4)$$

with $L$ the shell value in $R_J$. Combining Eqs. ((2)–(4)), with $ds^2 = dR^2 + R^2 d\theta^2$, we obtain

$$\frac{dB}{dt} = -\frac{3Bg(\theta)}{LR_J}v_{//} \qquad (5)$$

with

$$g(\theta) = \frac{\cos\theta}{\sin^2\theta}\frac{(3 + 5\cos^2\theta)}{(1 + 3\cos^2\theta)^{3/2}} \qquad (6)$$

Thus

$$v_{//}(f) = -\frac{df}{dt}\frac{LR_J}{3fg(\theta)} \qquad (7)$$

Finally, as observed drift rates are all negative, implying upgoing electrons, we can assume an adiabatic motion for the electrons. Following reflection at their mirror point close to the planet, the upgoing electron distribution possesses a loss-cone that can drive the ECM. The adiabatic hypothesis allows us to relate the parallel electron speed (energy) estimated at frequency $f$ to the total

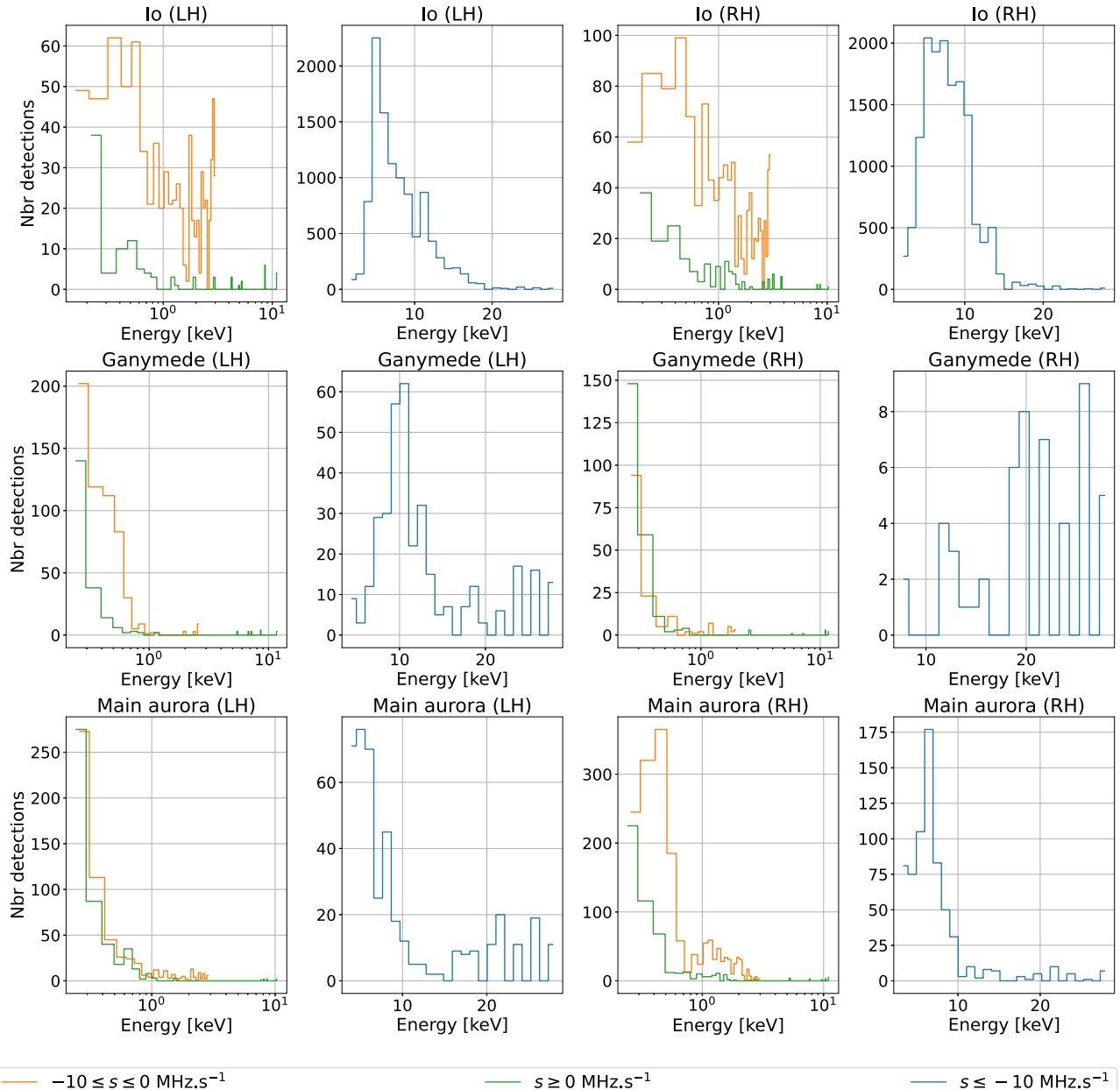

**Fig. 6 | Distribution of electron energies computed from the drift rate distributions of Fig. 3.** For each panel of Fig. 3, two panels are displayed here, with different scales: the left one shows the distributions of energies corresponding to $-10 \leq \mathrm{d}f/\mathrm{d}t < 0$ MHz/s (orange) and to positive drift rates (green), with energies on a logarithmic scale emphasizing values < 1 keV; the right one corresponds to $\mathrm{d}f/\mathrm{d}t < -10$ MHz/s (blue), with energies on a linear scale.

speed $v$, via the conservation of the perpendicular velocity down to the mirror point:

$$\frac{v_\perp^2}{f} = \frac{v^2 - v_{//}^2}{f} = \frac{v^2}{f_{\mathrm{mirror}}} \qquad (8)$$

hence

$$v^2 = \frac{v_{//}^2}{1 - \frac{f}{f_{\mathrm{mirror}}}} \qquad (9)$$

Finally, we assume that the frequency at the mirror point is the highest frequency of the Jovian decameter emission, $f_{\mathrm{mirror}} = 40$ MHz. Applying the above equations to all observed drift rates of Fig. 3, we obtain the distributions of electron energies displayed in Fig. 6.

Alternative possibilities, not considered here, involve the presence of an accelerating electric potential along the source field line, either distributed along the field line[76] or consisting of one or several potential drops[37,50].

## Data availability

The data analyzed are referenced and available at https://doi.org/10.25935/PBPE-BF82 (Lamy, L., G. Kenfack, P. Zarka, B. Cecconi, C. Viou, P. Renaud, F. Jacquet, A. Loh, L. Denis, A. Coffre: 2021. Nançay Decameter Array (NDA) Jupiter Juno-Nançay data collection (Version 1.0), [Data set], PADC (2021). The data obtained as a result of this analysis are available at https://doi.org/10.5281/zenodo.8098758,[77]. The derived data used to produce the figures in this study are provided in the Source Data file. Source data are provided with this paper.

## Code availability

The code used to process the data is available at https://doi.org/10.5281/zenodo.8098758, along with the code used to produce the figures[77].

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

## Acknowledgements

This article is based on observations made with the Nançay Decameter Array (NDA). The NDA is hosted by the Nançay Radio Observatory of the Observatoire de Paris (ORN, supported by CNRS, Université d'Orléans, OSUC, and Région Centre in France). We thank B. Cecconi and E. Thétas for their help in making the data available, and F. Jacquet (the NDA operator) for his daily data inspection work. P. Zarka acknowledges funding from the European Research Council (ERC) under the European Union's Horizon 2020 research and innovation program (grant agreement No 101020459 - Exoradio).

## Author contributions

E.M. and P.Z. developed the detection code, conducted the analysis and interpretation of the data and wrote the paper. L.L. provided insightful suggestions and comments (especially on the context, polarization and interpretation of multiple electron energies). S.H. helped with the physical interpretation of our results (Alfvénic acceleration of electrons).

## Competing interests

The authors declare no competing interest.
