## [Peer Review File · Nature Communications]

REVIEWER COMMENTS

Reviewer #1 (Remarks to the Author):

This is a fine paper that is definitely worthy of publication. My previous review provided what I believe are the noteworthy results and their significances. I do have comments, but I leave the resolution of these issues to the authors and the editors.

1) I understand that the Alfvén resonator theory is one that may lead other theories in explaining auroral acceleration in the literature. But, its viability is not assured. At Jupiter, whistler waves have been identified as possibly an important element in auroral processes (Elliott et al., 2020, Kurth et al. 2018). Solitary structures have been shown to be important at Earth (Ergun et al., 1998) and present at Jupiter (Sulaiman et al., 2022). Alfvén waves have been sought after and found over Jupiter’s main auroral regions but with powers much too weak to explain nominal auroral intensities (Lurch et al., 2022). I am happy with the authors to hypothesize the occurrence of the Alfvén resonator mechanism and to point to their own findings (and S-bursts in general) as supporting that hypothesis. But, treating that theory as having been proven, as the authors seem to do for Jupiter is not appropriate, I think.

2) The reader needs more hand-holding for the discussions of Figure 1. Where did the white boxes come from? I assume that in previous studies of general radio emissions, emissions that occur within these white boxes have been identified somehow as coming from Io or Ganymede. If so, the authors should state so more explicitly. I can understand the explanation of the background maps for the left panel (all Jovian radio emissions) but I do not understand the tersely described “Io induced” background for the right hand panel. Can the authors please explain? I understand that the pink symbols are identified as main aurora because they are not ordered by Io or Ganymede (or do not appear in the white boxes), but I would like the authors to state that fact more explicitly, and specifically to state that the authors cannot find or imagine any other source that might account for those emissions.

3) I had a previous comment that I reproduce here: “The modeling used to simulate the S-bursts in several Hess et al. papers is very impressive. However, it is unclear to me in my truly naïve reading the extent to which the “periodicity” is artificially introduced rather than accurately simulated. If it is artificially introduced, then saying (page 3 near the top) that “those showed quasi-periodic discrete bursts ---“would be misleading. The authors of this paper may be able to correct me. For me it is impressive enough to simulate the time evolution individual bursts.”

The authors responded that the periodicity was due to the “Alfvénic ionospheric resonator”. I well understood and understand the hypothesized source of the periodicity. My question is: was that resonator explanation simply assumed with the artificial introduction of a periodicity, or was the periodicity really a result of the hands-off simulation. My reading of the Hess et al. paper suggested that the periodicity was artificially introduced, but I could be wrong. If I am right, then the authors’ statement that “these bursts have been modeled end-to-end” is misleading (near line 28). I view the periodicity as the major evidence in support of the authors’ conclusions.

4) And once again, I find the statements regarding the observed Alfvén waves associated with Jupiter’s main aurora (near lines 188) to be stated in a misleading fashion. Yes, Alfvén waves are associated with the diffuse aurora. But, use of the word “especially” should be replaced with “only” unless the authors can provide evidence.

Reviewer #2 (Remarks to the Author):

This manuscript has been significantly improved from the previous version and is now basically acceptable for publication. I have one remaining concern, however. The electron speeds corresponding to the frequency drift are the order of tens of thousands km/s (e.g., a 5 keV electron has a speed of about 40,000 km/s). These velocities are in the same range as the Alfvén speed in the ionospheric Alfvén resonator (e.g., see Lysak et al., JGR Space Physics 2021, doi: 10.1029/2021JA029886). The authors should comment on this agreement; perhaps the similarities in velocity are relevant for the particle acceleration process.

Responses to the referees comments on the paper

« Discovery of Jovian radio bursts related to Ganymede and the main aurora, and implications on Alfvénic electron acceleration »

by E. Mauduit, P. Zarka, L. Lamy and S. Hess

Referees comments are in black, responses in blue. Line numbers refer to the revised manuscript with changes highlighted.

Referee #1 (Remarks to the Author):

This is a fine paper that is definitely worthy of publication. My previous review provided what I believe are the noteworthy results and their significances. I do have comments, but I leave the resolution of these issues to the authors and the editors.

Thank you.

1) I understand that the Alfvén resonator theory is one that may lead other theories in explaining auroral acceleration in the literature. But, its viability is not assured. At Jupiter, whistler waves have been identified as possibly an important element in auroral processes (Elliott et al., 2020, Kurth et al. 2018). Solitary structures have been shown to be important at Earth (Ergun et al., 1998) and present at Jupiter (Sulaiman et al., 2022). Alfvén waves have been sought after and found over Jupiter’s main auroral regions but with powers much too weak to explain nominal auroral intensities (Lurch et al., 2022). I am happy with the authors to hypothesize the occurrence of the Alfvén resonator mechanism and to point to their own findings (and S-bursts in general) as supporting that hypothesis. But, treating that theory as having been proven, as the authors seem to do for Jupiter is not appropriate, I think.

We now mention alternative auroral acceleration mechanisms in the penultimate paragraph (lines 219-222) of the main text. We cite there Kurth et al. (2018) [reference added] and Sulaiman et al. (2022). We cite Ergun et al. (1998) in the introduction (line 22). We do not cite Elliott et al. (2020) because they compute whistler waves (hiss) growth from observed MeV electrons rather than considering the role of whistler waves in electron acceleration (and the electrons involved in ECM have far lower energies, about 1-10 keV).

Note that the solitary structures reported by Sulaiman et al. (2022) were found only in Zone II, i.e. the downward current region, while the radio sources mostly lie above the diffuse aurora region, where Alfvén waves have been observed and intermittently diminish where 3–30 keV electron energy fluxes peak. Also, Lorch et al. 2022 (cited line 227) reported that Alfvénic turbulence within the mid-to-high latitude magnetosphere do have a Poynting flux sufficient to trigger auroral particle acceleration.

2) The reader needs more hand-holding for the discussions of Figure 1. Where did the white boxes come from? I assume that in previous studies of general radio emissions, emissions that occur within these white boxes have been identified somehow as coming from Io or Ganymede. If so, the authors should state so more explicitly. I can understand the explanation of the background maps for the left panel (all Jovian radio emissions) but I do not understand the tersely described “Io induced” background for the right hand panel. Can the authors please explain? I understand that the pink symbols are identified as main aurora because they are not ordered by Io or Ganymede (or do not appear in the white boxes), but I would like the authors to state that fact more explicitly, and specifically to state that the authors cannot find or imagine any other source that might account for those emissions.

We have rewritten the paragraph of the main text that describes and comments Figure 1 (lines 106-128), and the caption of that Figure, and hopefully it is clear now. We also have checked that the pink symbols are not related either to the Europa-Jupiter emission, so that an auroral (satellite-independent) remains the only interpretation we can reasonably think of.

3) I had a previous comment that I reproduce here: “The modeling used to simulate the S-bursts in several Hess et al. papers is very impressive. However, it is unclear to me in my truly naïve reading the extent to which the “periodicity” is artificially introduced rather than accurately simulated. If it is artificially introduced, then saying (page 3 near the top) that “those showed quasi-periodic discrete

bursts ---“would be misleading. The authors of this paper may be able to correct me. For me it is impressive enough to simulate the time evolution individual bursts.”

The authors responded that the periodicity was due to the “Alfvénic ionospheric resonator”. I well understood and understand the hypothesized source of the periodicity. My question is: was that resonator explanation simply assumed with the artificial introduction of a periodicity, or was the periodicity really a result of the hands-off simulation. My reading of the Hess et al. paper suggested that the periodicity was artificially introduced, but I could be wrong. If I am right, then the authors’ statement that “these bursts have been modeled end-to-end” is misleading (near line 28). I view the periodicity as the major evidence in support of the authors’ conclusions.

We have rewritten and hopefully clarified the part of the first paragraph of the main text describing past S-burst modelling work (lines 31-55). We recognize that the previous explanations were quite convoluted, with back and forth considerations on ECM generation of radio waves, Alfvén waves, electron acceleration and distribution function, burst discreteness and quasi-periodicity, and introduction of the ionospheric Alfvén resonator.

Now, we present the full chain of events in a linear order, hopefully making it more understandable and convincing. We tone down our expression of « end-to-end » modeling, rather explaining that there are several complementary modelling efforts that, combined together, provide a coherent generation scenario. We explain that what Hess et al. (2007) showed is that the period of the Alfvén wave introduced in the simulation (5 Hz, close to that predicted to exit the resonator by Su et al., 2006) is found to modulate the electron's distribution function as well as the ECM radio growth rate. We note that the discreteness and quasi-periodicity of S-bursts naturally emerge – as explained in Hess et al. (2007) – from the fact that a « bunch » of electrons is accelerated (and escapes) downwards every other half- Alfvén wavelength from the lower edge of the high E_{\parallel} region.

4) And once again, I find the statements regarding the observed Alfvén waves associated with Jupiter’s main aurora (near lines 188) to be stated in a misleading fashion. Yes, Alfvén waves are associated with the diffuse aurora. But, use of the word “especially” should be replaced with “only” unless the authors can provide evidence.

Our word « especially » came from the fact that Sulaiman et al. (2022) stated : « *Alfvénic fluctuations are most prominent in the diffuse aurora and are repeatedly found to diminish in Zone-I and Zone-II, likely due to dissipation, at higher altitudes, to energize auroral electrons.* » But we agree that this is only one possible interpretation, thus we removed the word « especially » (line 232), without emphasizing with « only ». We hope that this is acceptable.

Referee #2 (Remarks to the Author):

This manuscript has been significantly improved from the previous version and is now basically acceptable for publication. I have one remaining concern, however. The electron speeds corresponding to the frequency drift are the order of tens of thousands km/s (e.g., a 5 keV electron has a speed of about 40,000 km/s). These velocities are in the same range as the Alfvén speed in the ionospheric Alfvén resonator (e.g., see Lysak et al., JGR Space Physics 2021, doi: 10.1029/2021JA029886). The authors should comment on this agreement; perhaps the similarities in velocity are relevant for the particle acceleration process.

As explained in Hess et al. (2007), the Alfvén velocity at the altitude where acceleration occurs is much larger than the velocity of 1-10 keV electrons, thus the interaction is non-resonant. Acceleration up to a few keV occurs in both directions (depending on the sign of E_{\parallel}) in half an Alfvén wavelength, whatever the direction of propagation of the Alfvén wave. Thus the similarity of the Alfvén velocity inside the ionospheric resonator and of the electrons velocity in the radio sources is a coincidence.

We now precise in the text (lines 40-45) : « ... *these waves give rise to parallel bi-directional electric fields over a broad range of altitude along the IFT ; at the lower edge of this range, 0.5-1 R_J (= Jupiter's radius = 71492 km) above Jupiter's ionosphere, these parallel electric fields were shown to be able to accelerate downwards electrons up to 1-10 keV over half an Alfvén wavelength ...* »

REVIEWERS' COMMENTS

Reviewer #1 (Remarks to the Author):

The responses of the authors to my criticisms are satisfactory, and I remove all of my objections to publication.

Reviewer #2 (Remarks to the Author):

The authors have responded positively to my previous comments and, in my opinion, to the comments of the other reviewer. I now recommend the article for publication.